# Framework nucleic acids as programmable carrier for transdermal drug delivery

Christian Wiraja[1], Ying Zhu[2], Daniel Chin Shiuan Lio[1,3], David C. Yeo[1], Mo Xie[2], Weina Fang[4], Qian Li[4], Mengjia Zheng[1], Maurice Van Steensel [5], Lihua Wang[2,6], Chunhai Fan[4] & Chenjie Xu[1,3,7]

DNA nanostructures are promising drug carriers with their intrinsic biocompatibility, uniformity and versatility. However, rapid serum disintegration leads to low bioavailability at targeted sites following systemic administration, hindering their biomedical applications. Here we demonstrate transdermal delivery of framework nucleic acids (FNAs) through topical applications. By designing FNAs with distinct shapes and sizes, we interrogate their penetration on mice and human skin explant. Skin histology reveals size-dependent penetration, with FNAs ≤75 nm effectively reaching dermis layer. 17 nm-tetrahedral FNAs show greatest penetration to 350 μm from skin periphery. Importantly, structural integrity is maintained during the skin penetration. Employing a mouse melanoma model, topical application of doxorubicin-loaded FNAs accommodates ≥2-fold improvement in drug accumulation and tumor inhibition relative to topically-applied free doxorubicin, or doxorubicin loaded in liposomes and polymeric nanoparticles. Programmable penetration with minimal systemic biodistribution underlines FNA potential as localized transdermal drug delivery carriers.

[1] School of Chemical and Biomedical Engineering, Nanyang Technological University, 62 Nanyang Drive, Singapore 637459, Singapore. [2] Division of Physical Biology and Bioimaging Center, Shanghai Synchrotron Radiation Facility, CAS Key Laboratory of Interfacial Physics and Technology, Shanghai Institute of Applied Physics, Chinese Academy of Sciences, Shanghai 201800, China. [3] NTU-Northwestern Institute for Nanomedicine, Nanyang Technological University, 50 Nanyang Avenue, Singapore 639798, Singapore. [4] School of Chemistry and Chemical Engineering, and Institute of Molecular Medicine, Renji Hospital, School of Medicine, Shanghai Jiao Tong University, Shanghai 200240, China. [5] Lee Kong Chian School of Medicine, Nanyang Technological University, Singapore 639798, Singapore. [6] Shanghai Key Laboratory of Green Chemistry and Chemical Processes, School of Chemistry and Molecular Engineering, East China Normal University, 500 Dongchuan Road, Shanghai 200241, China. [7] National Dental Centre of Singapore, 5 Second Hospital Ave, Singapore 168938, Singapore. These authors contributed equally: Christian Wiraja, Ying Zhu. Correspondence and requests for materials should be addressed to C.F. (email: fanchunhai@sjtu.edu.cn) or to C.X. (email: cjxu@ntu.edu.sg)

DNA origami is an emerging technique that exploits programmable folding of single-stranded DNA molecules to generate complex framework nucleic acids (FNAs) ranging from periodic arrays to three-dimensional (3D) architectures[1]. DNA can self-assemble into arbitrary shapes with full addressability, allowing the fabrication of nanostructures with well-defined form and precisely arranged heteroelements[2]. Given their biocompatibility, versatility, and controllable size and shape, FNAs hold great promise for drug delivery[3,4]. However, DNA origami based in vivo application has predominantly been explored through invasive needle injection. In such systemic delivery, DNA structures are subjected to rapid disintegration and digestion besides immune- and renal-clearance, leading to low bioavailability at targeted sites[5,6]. In mouse model, their half-life time in circulation is only of minutes or up to a few hours after chemical modification of DNA backbones[2,6,7], forming a major barrier for widespread application of DNA origami-based nanomedicine.

Transdermal drug delivery (TDD) is an attractive mean of drug administration, given its non/minimally invasive nature, high-patient compliance, and direct route of entry bypassing gastro-intestinal or liver metabolism. Consequently, intended treatment can be achieved with minimal side effects[8,9]. Studies have revealed that nanoparticles (NPs) of appropriate size, shape, and surface properties can reach viable epidermis or even dermis layer for potent therapy in vivo[10,11]. One-specific case is the skin penetration of spherical nucleic acids (SNAs), gold NPs coated with tightly organized oligonucleotides. Without assistance from external devices, these SNAs can penetrate through the stratum corneum and epidermis barrier, to reach the dermis region[11,12]. Although the mechanism remains mostly unclear, the similarity of size and surface chemistry between FNAs and SNAs leads us to hypothesize that FNAs would penetrate the skin as well. If successful, this attempt would provide an alternative carrier for TDD with precisely controlled size, shape and surface chemistry, while concurrently open new application route to circumvent the susceptibilities of FNAs in systemic drug delivery (Fig. 1). Skin diseases like melanoma, abnormal scarring, and atopic dermatitis[13,14] would benefit from the resulted formulations such as FNAs containing RNA interference moieties and/or antisense oligonucleotides[15].

This study explores the skin penetration ability of FNA structures between 20 and 200 nm in size. We observes a size-dependent skin penetration phenomenon, with 20 nm tetrahedron FNA (TH) reaching into the dermis region. FNA integrity was well maintained during the skin penetration. Taking the optimized system, we loaded 20 nm TH structures with doxorubicin (DOX) through intercalation for the topical treatment of melanoma tumor in mice. Compared with other topical carriers (i.e. liposomes and polymeric NPs), we observe twofold enhancements in DOX delivery and tumor inhibition, corroborating FNA efficacy in achieving TDD.

## Results

**In vitro characterization of FNAs**. To examine the topical penetration and drug carrier capability of well-defined FNAs, we have designed 8 types of FNAs including 3 tetrahedrons, 2 cylindrical rods, 2 rectangles, and 1 triangle (Fig. 2a), ranging from 81.9 to 4711.9 kDa in size (Supporting Table 1). FNAs were synthesized through self-assembly via base pairing hybridization, assisted by the presence of divalent cations (e.g. $Mg^{2+}$) to minimize electrostatic repulsion from the negatively charged nucleotides.

FNAs were examined through atomic force microscopy (AFM) or gel electrophoresis. AFM confirmed the desired assembly of large FNAs, including tetrahedron with 337 base

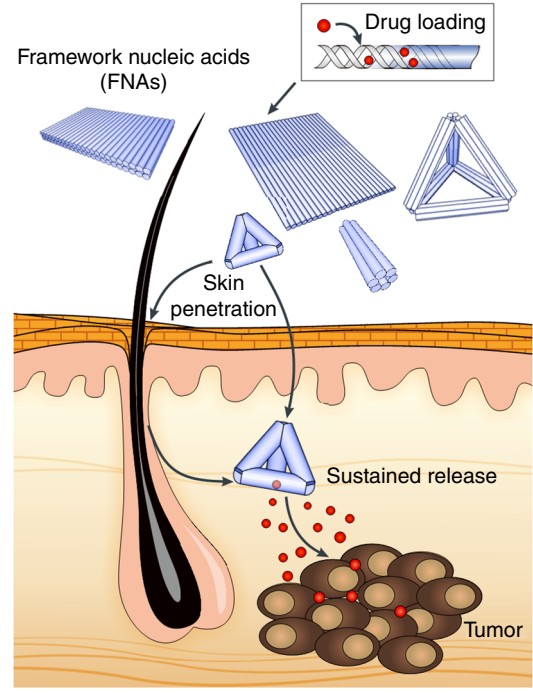

**Fig. 1** Schematic representation of the topical application of framework nucleic acids (FNA) for transdermal drug delivery. Upon successful penetration, FNAs can release conjugated/intercalated drugs, including but not limited to melanoma treatment

pairs (bps) (TH337), 6-helical rods with 14,498 bps (6H14498), rectangular plane with 13,730 bps (R13730), rectangular box with 14,498 bps (B14498) and triangular plane with 14,498 bps (T14498) (Supporting Fig. 1A). Tetrahedron with 21 bps (TH21), tetrahedron with 37 bps (TH37) and 6-helical rods with 714 bps (6H714) were too small for AFM imaging and were confirmed using gel electrophoresis. As shown in Supporting Fig. 1B, TH21 migrated the furthest in the gel, matching the 220 bps band from the DNA ladder, followed by 6H714 and TH37 which migrated less distance from the 300 bps band. Interestingly, 6H714 migrated slightly further than TH37 despite its larger molecular weight, which might be due to its rod-like shape[16]. Finally, dynamic light scattering examination revealed their hydrodynamic diameters ($D_h$; Supporting Fig. 1C & D). The sizes of all tetrahedron structures have gaussian distribution with the average $D_h$ of 17 nm for TH21, 44 nm for TH37 and 187 nm for TH337. Both 6-helix (6H714 and 14498) structures have two peaks, likely corresponding to the width (~30 nm) and length of the rods (~66/ 220 nm). Meanwhile, the remaining large rectangular or triangular nanostructures (R13730, B14498, and T14498) have a similar average $D_h$ between 140 and 170 nm. Notwithstanding slight alteration due to shape differences, average $D_h$ increased in correlation with the molecular weight of FNAs (Supporting Fig. 1E).

We then examined interaction of these FNAs with skin cells in vitro. First, FNA internalization kinetic was evaluated using smallest Cy5.5-tagged TH21 on skin fibroblast cells (Supporting Fig. 2A & B). There were ~18 and 50-fold increases in cell fluorescence after 6 and 24-h incubation respectively. Intracellular localization of FNAs was revealed through confocal imaging. As shown in Supporting Fig. 2C, TH21 signal (Cy5.5-tagged; red) localized in the cytoplasm (i.e. outside of Hoechst33342-stained nuclei; blue) and partially overlapped with Lysotracker signal

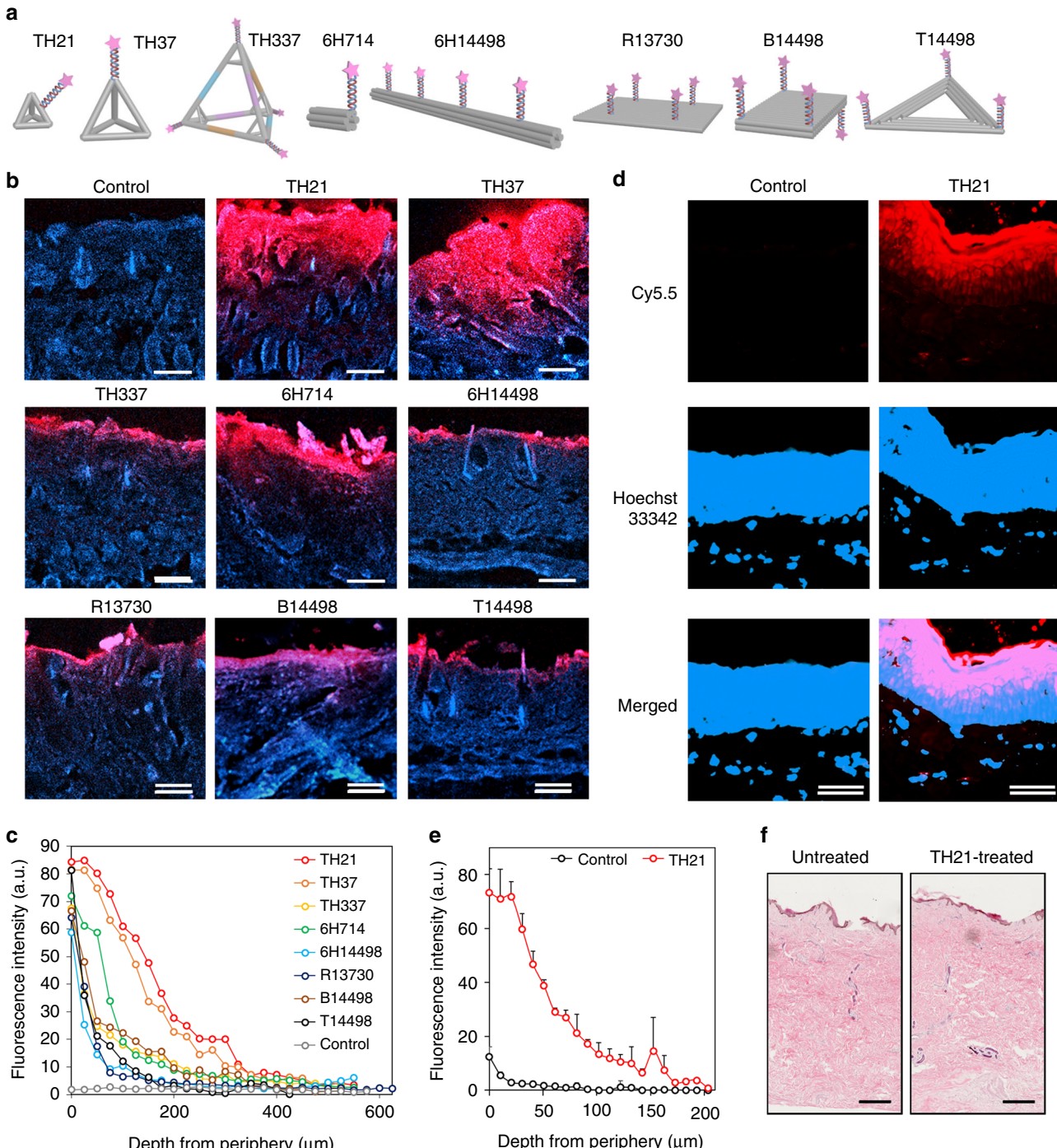

**Fig. 2** Topical delivery and penetration of FNAs through mice and human skin: **a** Schematic illustration of 8 kinds of FNAs (not drawn to scale). **b** Representative fluorescence images of mouse skin histology post various FNAs penetration (24 h; red: Cy5.5-tagged FNAs, blue: Hoechst33342-stained skin); **c** profile of Cy5.5 fluorescence signal from FNAs along skin depth in **b**. **d** Representative fluorescence images of human skin explant histology post TH21 FNA penetration (24 h; red: Cy5.5-tagged FNAs, blue: Hoechst33342-stained skin); **e** profile of fluorescence signal from Cy5.5-TH21 FNA along human skin depth in **d**. **f** Hematoxylin & eosin staining showing TH21 treatment minimally affects human skin morphology (left: untreated skin, right: TH21-treated skin). Source data are provided as a Source Data file. Scale bar: 200 μm. Error bars represent standard deviation

(staining endo-lysosomes; green), confirming that FNAs entered the cells through the endolysosomal pathway[17].

Comparing internalization of all FNAs after 24-h incubation, TH21 showed greatest intracellular accumulation, followed by TH37 and 6H714 (Supporting Fig. 3 and Supporting Fig. 4A). The remaining structures had neglectable cellular uptake post signal normalization. This aligns well with previous reports in which there was better cellular internalization of compact, low-aspect ratio FNAs in the size of 50–80 nm[18]. Scavenger receptors, facilitating cellular uptake of DNA moieties, obviously prefer to bind and transport FNAs in this size range[19]. All FNAs showed minimal influence on the viability of the skin fibroblasts and keratinocytes, even at the high concentration of 1 μg/mL (Supporting Fig. 4B). In fact, most of them promoted cell proliferation, presumably by providing useful bioproducts following its catabolism[20].

**Topical delivery of FNAs in mouse and human skin explant.**
Eight kinds of FNAs were mixed with the moisturizer (i.e.
Aquaphor®) under the same mass concentration (100 µg/ml). To
adjust for molecular weight differences, one fluorophore was
tagged on each TH21, TH37, and 6H714, 3 fluorophores on
T14498 and 4 on each TH337, 6H14498, R13730, and B14498
(Supporting Table 1). In ensuring successful tagging and main-
tenance of purity, excess fluorophore strands were added during
assembly of large FNAs, which were subsequently removed
through repeated chromatography or centrifuge filtrations (Sup-
porting Table 2–Supporting Table 9). Overall, comparable
fluorescence level (within an order) was observed from the FNAs
at the same mass concentration (Supporting Fig. 5). The for-
mulations were topically applied on the back of mice and covered
with Tegaderm™ dressing for 24 h (Supporting Fig. 6A). After
wiping off excess cream, the mice were imaged with in vivo
imaging system (IVIS). Finally, mice were killed, and treated skin
regions were taken for histological analysis.

IVIS showed skin treated with TH21, TH37, and 6H714 had
greater fluorescence signal than skin samples treated with the other
5 FNA structures (Supporting Fig. 6B). Skin histology revealed
significant penetration by TH21, TH37, and 6H714 among others
(Fig. 2b, c). Fluorescence signal of 6H714 remained strong until
~100 µm deep, signifying noteworthy penetration through the
epidermis and up to early dermis region. Meanwhile, even more
remarkable entry was noted for TH37 and TH21, reaching ~350 µm
beneath the skin surface. Larger structures (TH337, 6H14498,
R13730, B14498, and T14498) were mostly retained only at the
epidermis region (~50–75 µm from the skin surface).

Upon closer examination through keratin14 (Krt14) counter-
staining, we noted that FNAs localize in deep dermis region
predominantly around the hair follicles and sweat glands (Support-
ing Fig. 7A, white arrow). This suggests that trans-appendageal route
played a crucial role for extensive FNA skin penetration, in
agreement with reported transdermal study with liposomes or
polymeric NPs[10,21]. It is noteworthy that FNA penetration until
epidermis-dermis boundary was also observed at area without
follicles or glands, suggesting partial role of trans/intercellular route
in enabling topical delivery of FNAs (yellow arrow). By scanning
and correlating fluorescence signals of FNA and Krt14-stain, we
estimate 20–25% internalization of FNAs occurring via non-trans-
appendageal route (Supporting Fig. 7B & C).

Taking TH21 as a proof-of-concept, the FNAs were addition-
ally tested on human skin explant for their penetration ability and
biocompatibility. As shown in Supporting Figure 8 & Fig. 2d (for
×40 and ×200 imaging magnification, respectively) and quantified
in Fig. 2e, Cy5-tagged TH21 formulation (red) penetrated
significantly through the epidermis, reaching into the dermis
region (Hoechst33342 signal: blue). Lack of red fluorescence from
control group (PBS + cream) confirmed minimal fluorescence
signal from the cream itself. TH21 signal reached ~160–170 µm
beneath the skin surface, around the stratum basale and into the
dermis region on intact human skin. Noting the more robust,
denser barrier of the human skin, FNA penetration could be
enhanced with assistive technology (e.g. microneedles, iontophor-
esis) to enable deeper delivery into the dermis region[22,23].
Nevertheless, FNAs remain a promising tunable TDD nanocar-
rier, with certain extent of penetration and minimal alteration/
damage-induced towards skin morphology during its delivery
(Fig. 2f).

Beyond the extent of skin penetration, integrity of FNAs
during this process is also crucial for TDD application. As proof-
of-concept, we examine this issue using symmetrical TH21. To
see whether the stability of one TH21 edge can represent the
stability of all edges, we ran gel electrophoresis study on all
permutations of TH21 chain assembly (Fig. 3a). As shown in

Fig. 3b, similar distances were traveled for all permutations
containing same number of ssDNA chains (e.g. missing 2 chains:
2A-2F, missing 1 chain: 3A-3D). This, supported with equivalent
A/T and C/G ratio of each chain, substantiates our hypothesis
regarding uniformly-stable TH21. We then designed a dual
(Cy3/Cy5)-tagged TH21 structures without or with BHQ3
quencher (TH or TH', respectively; Fig. 3c and Supporting
Table 10). Due to the proximity of BHQ3 with Cy5 fluorophore
in TH', the structure had significantly lesser Cy5 fluorescence
compared to unquenched TH, while retaining similar Cy3
fluorescence. In vitro, Cy5/Cy3 signal of TH' was 35% of that of
TH. Meanwhile, incomplete TH' (i-TH', lacking one side to
mimic degraded TH') exhibited recovered Cy5 fluorescence
(Cy5/Cy3 ratio of 98% of that of TH; Fig. 3d). In this manner,
Cy5/Cy3 fluorescence ratio following topical penetration of TH'
can be compared to that of unquenched TH, to evaluate the
extent of disintegration of the TH' structure. Following the
topical application on mice, TH' showed lower Cy5/Cy3 signal
than TH from both IVIS (Fig. 3e) and fluorescence imaging of
skin histology (Fig. 3f, g). Under IVIS imaging, the Cy5/Cy3 ratio
of TH' was 44% of that of TH (Fig. 3e). The slight disparity from
the in vitro result might be due to variability in mouse skin
condition. This is supported by histological result, which showed
an average of 48% for Cy5/Cy3 ratio from TH' group to TH
group, across its penetration depth (Fig. 3f, g). Significant Cy5
fluorescence of the quenched TH' group at the skin surface is
likely due to signal saturation from the high FNA concentration
applied. Overall, these results suggest that integrity of the
tetrahedron structure was well-retained during the skin penetra-
tion process.

**Transdermal drug delivery with topical FNAs.** We finally
explored the role of FNAs as drug carrier in transdermal delivery.
As a proof-of-concept, we loaded DOX into TH21 (TH-DOX)[24,25].
Slower migration rate of TH-DOX on gel electrophoresis as
compared to Cy5.5-tagged TH or TH only indicated successful
DOX loading (Supporting Figure 9A). Moreover, fluorescence
measurement of TH-DOX solution against solutions of known
DOX concentrations allowed the precise quantification of DOX
amount, ~0.57 µg DOX for 100 µg of TH21 or 49.72 nmol DOX
per 1 nmol of TH21. When these results were normalized with the
number of nucleotide (nt) in the TH21 (252 nt, i.e. ~0.2 DOX/nt),
this loading efficiency is comparable to previously-reported 1–9
µM DOX loading in 1 nmol of ~9000 nt DNA nanotube (i.e.
~0.1–1 DOX/nt)[25]. Performing gel electrophoresis study on
unloaded (TH21) or loaded TH-DOX post distinct incubation
period in 10% serum condition, the presence of intercalated DOX
significantly extended TH lifespan from enzymatic degradation
(half-life of >48 h than 26 h on unloaded TH, Supporting Fig-
ure 10). Thus, while TH21 can be expected to remain effective as
carrier within 1–2 days, TH-DOX formulation can remain effica-
cious for even longer.

The prepared TH-DOX had a 72-h inhibitory concentration
(IC50) of 153.45 nM against B16F10 mice melanoma cells, more
efficient than free DOX (IC$_{50}$: 335.81 nM; Fig. 4b). Such
observation is consistent to previous report on breast cancer
cells (MDA-MB231, MDA-MB468, and MCF7), where lower
DOX IC$_{50}$ value was reported following DNA nanostructures
intercalation and intracellular delivery[24,25]. Possible explanations
for this are related to disruption of endo-lysosomal pathway
(through prevention of acidification) and sustained release of
DOX from FNAs, leading to greater cellular distribution and
overall availability[24].

To better assess the performance of TH-DOX as TDD carrier,
we also synthesized commonly adopted topical drug carriers in

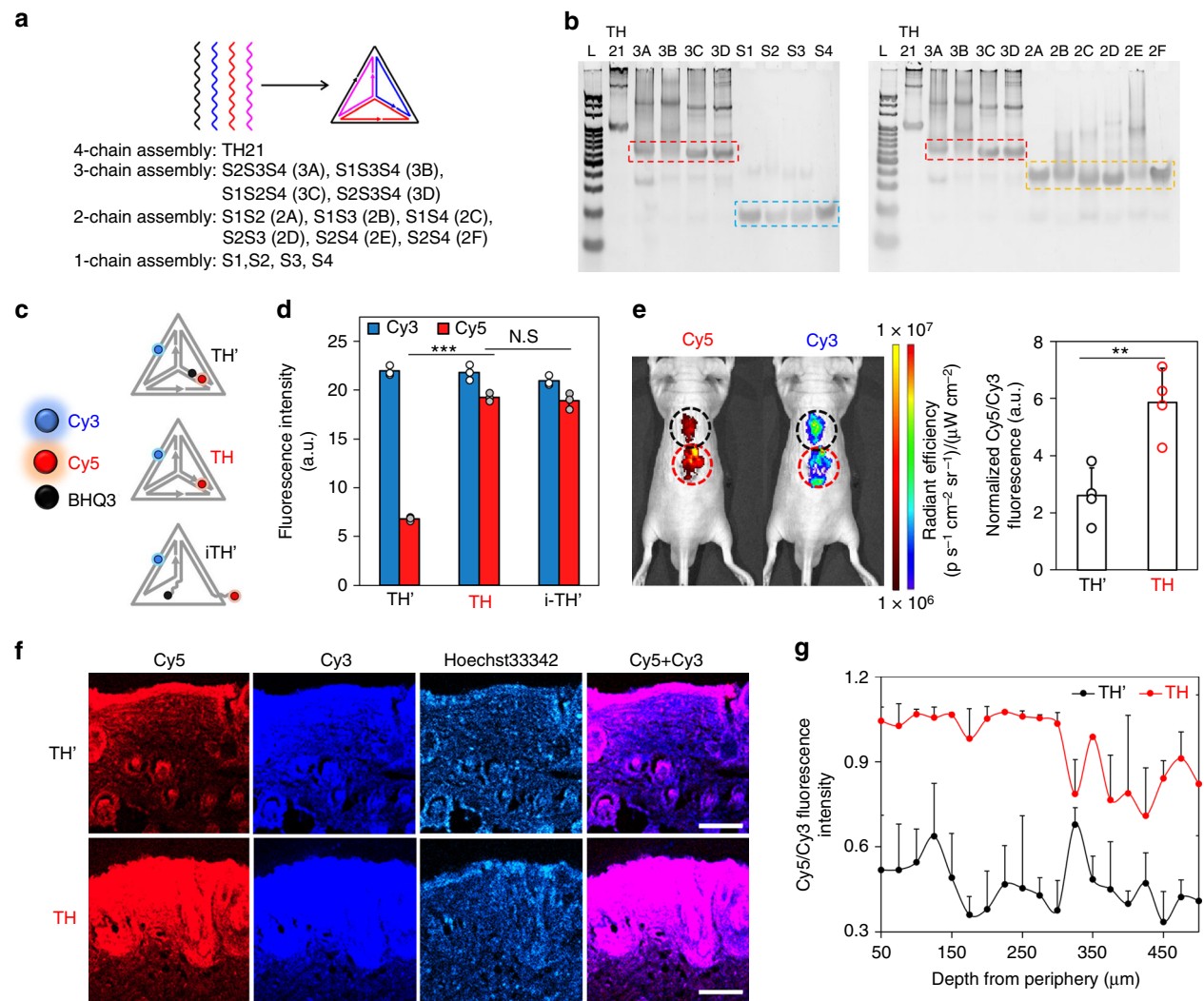

**Fig. 3** Assessing integrity of FNAs during their skin penetration: **a** All assembly permutations from 4 TH21 component chains (S1, S2, S3, S4). **b** Gel electrophoresis study of all assembly in a (L: 10 bp-ladder, dotted boxes show similar distance traveled for assemblies with 1, 2, or 3 chains). **c** Design of dual-tagged quenched TH', unquenched TH and incomplete TH' (i-TH'). **d** Cy3 and Cy5 fluorescence intensities of TH', TH and i-TH' (Blue and red column, respectively). Close proximity of BHQ3 to Cy5 fluorophore results in low Cy5 yield in intact TH'. Topical application of TH' and TH on mice: **e** IVIS in vivo imaging and quantification of Cy3 and Cy5 intensity ratio (black circle: TH', red circle: TH); **f** fluorescence imaging of histological sections (red: Cy5, dark blue: Cy3, light blue: Hoechst33342, and merged: Cy5&Cy3 channel) and **g** the related signal quantification ratio (Cy5/Cy3) of treated skin (black line: TH', red line: TH). Each dot on bar charts represents independent sample measurement. Source data are provided as a Source Data file. Scale bar: 200 µm. Error bars represent standard deviation. N.S represents non-significance, **$p < 0.01$ and ***$p < 0.001$ (one-way ANOVA)

liposomes and degradable polymeric (i.e. poly(lactic-co-glycolic acid)/PLGA) NPs as comparison[26,27]. The drug loading efficiencies in these larger (~100 nm) NPs were ~2-folds of TH nanostructures (Fig. 4a). Upon longer term storage (i.e. 1 week at 4 °C), however, PLGA-DOX and LIP-DOX showed significant drug leakage (19.3 and 10.8%; Fig. 4c). Moreover, LIP-DOX are limited with tendency to aggregate upon storage (Fig. 4d). We then investigate skin penetration ability of TH-DOX against these carriers, free DOX, free DOX following microneedle (MN) application, and DOX-intercalated 21-bp dsDNA segments (DS-DOX). To note, drug loading was similar between DS21 and TH21 (Fig. 4a). In particular, solid poly(methyl methacrylate) (PMMA) MN (600 µm tall) was finger-pressed for 1 min to disrupt skin membrane integrity (Fig. 4e). As represented and quantified in Fig. 4f, g, TH-DOX facilitated deep penetration in pig skin reaching ~400 µm from periphery, comparable to when

free DOX could diffuse following PMMA MN pre-application. LIP-DOX reached ~250–300 µm depth from periphery. DS-DOX (~8 nm in hydrodynamic diameter) showed significantly less penetration than TH-DOX, reaching 150 µm from periphery. Likely, they were retained by the initial few cell layers through interaction with membrane receptors. Coupled with its susceptibility to degradation, they were consequently unable to maintain integrity to penetrate and reach deeper skin layers. PLGA-DOX penetrated minimally to a depth of 100 µm, comparable to that achieved by free DOX.

**Efficacy of topical FNAs for cancer treatment**. TH-DOX formulation was later topically applied on the mouse melanoma model, carrying ~6 mm-sized subcutaneous melanoma tumor in both hind legs (Fig. 5a). Following 24 h treatment and removal of excess cream, IVIS revealed threefold DOX delivery at tumor

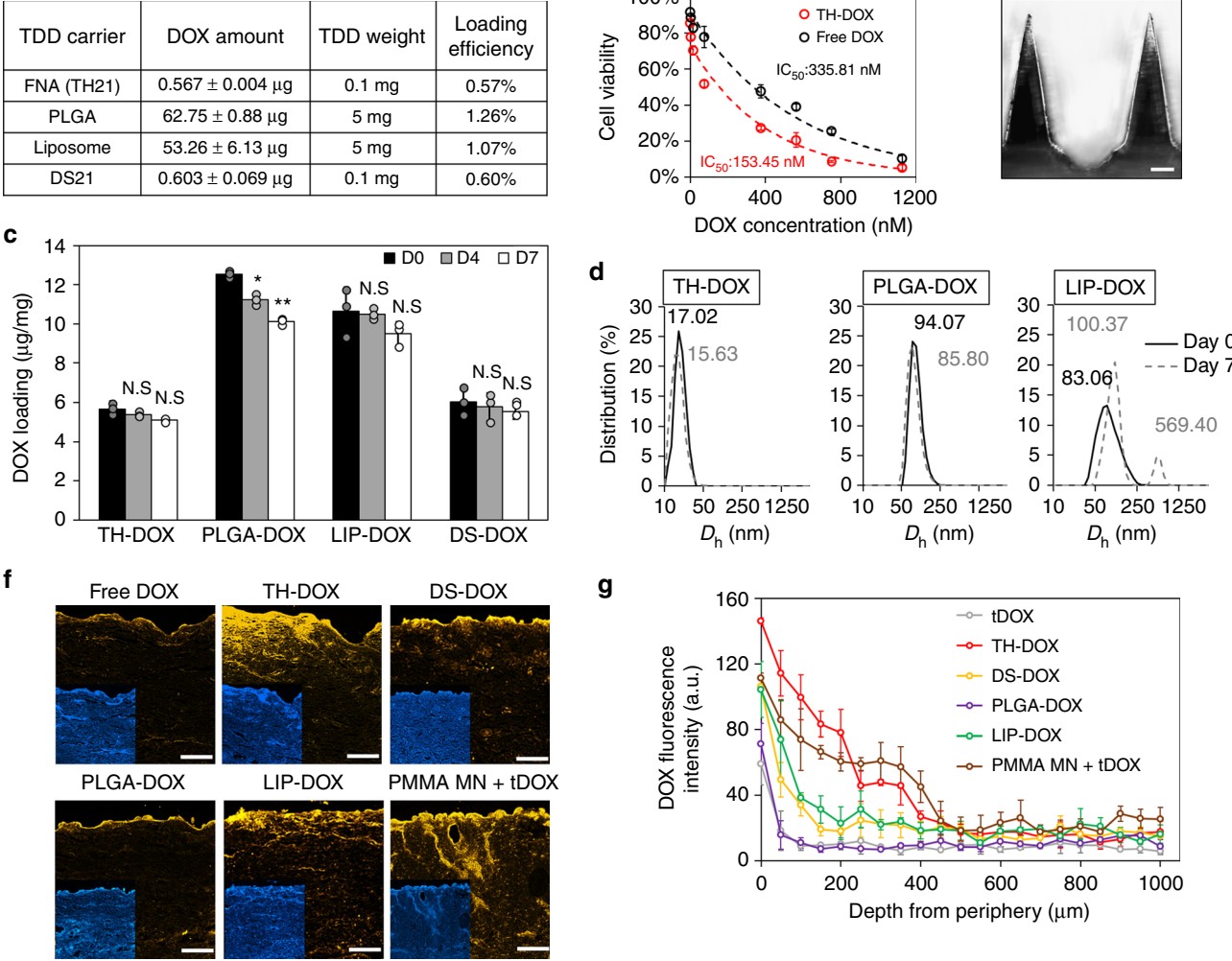

**Fig. 4** Assessing FNA as transdermal drug delivery carrier: **a** Doxorubicin (DOX) loading efficiency of FNA, 21 bp dsDNA segments (DS21), PLGA NP and liposomes. **b** Inhibitory concentration (IC50) assay of TH-DOX (red) and free DOX (black) after 72 h of incubation. **c** Loading stability (day 0: black, day 4: gray and day 7: white) and **d** physical stability of TH-, DS-, PLGA-, and LIP-DOX upon storage at 4 °C. Solid lines: size distribution at day 0, dotted lines: day 7 post storage. Each dot on bar chart represents independent sample measurement. **e** Morphology of solid PMMA microneedle for membrane disruption as control. **f** Representative images and **g** signal quantification of DOX fluorescence following carrier penetration in pig skin for 24 h. Yellow channel: DOX fluorescence; inlet: Hoechst33342-stained skin. Source data are provided as a Source Data file. Scale bar: 200 μm. Error bars represent standard deviation. N.S represents non-significance, *$p < 0.05$ and **$p < 0.01$ (one-way ANOVA)

region following TH21 application than free DOX application (Fig. 5b). This observation is consistent over six-independent experiments (Fig. 5c). Histology further shows that TH21 brought DOX as deep as 400–450 μm from the skin surface, significantly deeper than the mouse skin covering the tumor area (Fig. 5d, Supporting Figure 11). The mice skin here is notably thinner (100–125 μm) than physiological skin thickness (≥350 μm), presumably from the stretching pressure exerted by underlying tumor. DOX signal decreased along increasing skin/tumor depth (Fig. 5f), matching that of TH21 (Fig. 5e). In comparison, topically delivered free DOX reached only ~50–75 μm beneath the skin surface, mainly retained at the skin without entering the tumor (Fig. 5f, Supporting Figure 11). Through DOX signal quantification at the subcutaneous tumor region (i.e. signal from 100 μm depth onwards), a 5.67-fold increase in DOX accumulation was achieved via TH-DOX delivery, as compared to free DOX treatment (Fig. 5g). These results suggest the efficacy of TH-DOX to deliver drugs to subcutaneous tumor site.

To corroborate the successful FNA-facilitated DOX delivery to subcutaneous tumor, we further evaluated its effectiveness in inhibiting tumor growth. As depicted in Fig. 6a, we extended the study 2 weeks after the first application of TH-DOX, with three topical treatments a week and bi-daily recording of tumor volume. For comparison purposes, the same amount of DOX was topically delivered by itself (tDOX), within DS-DOX, PLGA NPs (PLGA-DOX), liposomes (LIP-DOX) or through intra-tumoral injection (iDOX) under the same frequency (Fig. 6a). As shown in Fig. 6b and Supporting Figure 12, TH-DOX treatment significantly hindered the tumor growth over other topically applied DOX carrier (DS-DOX, PLGA-DOX, LIP-DOX). At the final time point (day 17 post injection), tumor size on mice treated with TH-DOX was ~33% of the untreated controls (544 mm³ relative to 1653 mm³; Fig. 6c). This result was most comparable to that achieved through PMMA MNs (397 mm³, 24% of the untreated tumors; Supporting Figure 13A). In contrast, DS-DOX, LIP-DOX, PLGA-DOX, and tDOX treatment had less inhibitory effect on tumor

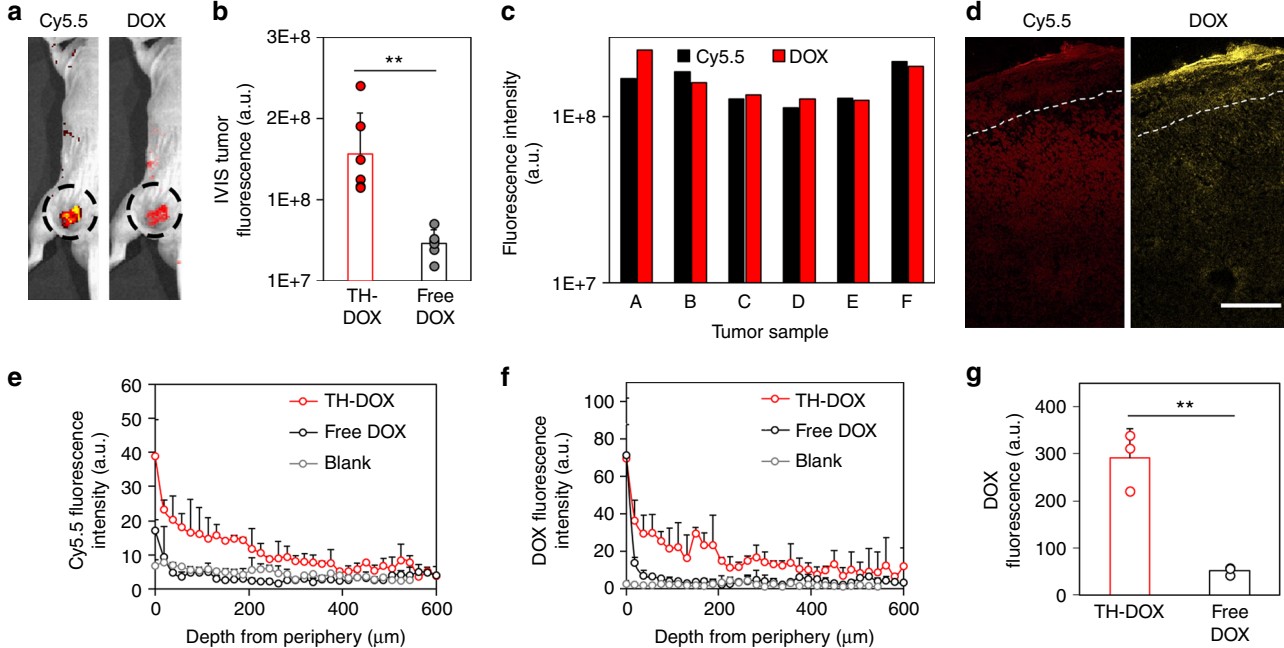

**Fig. 5** Topical delivery of DOX and TH-DOX to melanoma tumor model in mouse: **a** Representative in vivo imaging post the topical application of TH-DOX showing co-localized Cy5.5 TH and DOX signal (dotted region); **b** Quantification of DOX fluorescence at tumor site from in vivo imaging, following TH-DOX (red) or free DOX (black) application; **c** Consistent tumoral DOX delivery by TH-DOX with comparable Cy5.5-FNA (black) and DOX (red) signal across six-independent tumors. **d** Representative fluorescence imaging of histological sections following TH-DOX application (red channel: Cy5.5-TH, yellow channel: DOX, dotted line marks skin/tumor boundary). Corresponding quantification of **e** Cy5.5-TH signal and **f** DOX signal in skin and tumor (red line: TH-DOX treated, black line: free DOX treated, gray line: blank control). **g** Quantification of the DOX fluorescence in tumor area post the topical delivery of TH-DOX (red) and free-DOX (black). Each dot on bar charts represent independent sample measurement. Source data are provided as a Source Data file. Scale bar: 200 μm, Error bars represent standard deviation. **$p < 0.01$ (one-way ANOVA)

growth, with 68.0, 53.9, 74.5, and 84.5% of the untreated tumors, respectively (Fig. 6d). Notably, TH only group showed negligible inhibition compared to untreated tumors (Fig. 6b, d).

The isolated tumor tissues were further examined with IVIS and histology. IVIS imaging revealed that repeated tDOX and PLGA-DOX treatment resulted in minimal DOX fluorescence over untreated tumor, likely due to insufficient penetration. In comparison, TH-DOX and iDOX, as well as PMMA MN+tDOX significantly enhances DOX content (Fig. 6e and Supporting Figure 13B). By homogenizing these tumors in PBS and plotting it against a standard curve, we found that every gram of TH-DOX and iDOX-treated tumors contained ~7.1 and 8.5 μg of DOX, significantly >1.4 μg found in tDOX-treated tumors, 2 μg in PLGA-DOX, 3.3 μg in DS-DOX and 4.3 μg in LIP-DOX tumors (Fig. 6f and Supporting Figure 13C). TH-DOX facilitated 5.1-fold accumulation of DOX as compared to tDOX, matching results from fluorescence quantification in Fig. 5g. By imaging sliced tumor samples under confocal microscopy, we confirmed the presence of considerable amount of DOX following TH-DOX, PMMA MN+tDOX and iDOX treatment (Fig. 6g and S14), with concomitant increase in Annexin V expression (indicative of apoptotic tissue; Fig. 6h and Supporting Figure 14). Overall, these results demonstrated efficacious transdermal DOX treatment through FNA delivery, validating its effective penetration.

**Safety of FNAs for localized transdermal drug delivery.** Finally, we examined the accumulation of DOX in the major tissues/ organs following different delivery formulations. As shown in Fig. 6b, intra-tumoral DOX injection (iDOX) led to greatest efficacy in inhibiting tumor growth. However, it also resulted in the largest DOX accumulation on other organs of the mouse (i.e.

heart, lung, liver, spleen, and kidney; Fig. 7a, b). There was even greater DOX accumulation on the liver than the tumor itself (Fig. 7b). Such result was reflected by the amount of DOX contained in the blood serum (Fig. 7c, d), where serum from iDOX group contained >2.4-fold DOX as compared to that from other DOX carrier. Of note, TH-DOX presented low-serum DOX concentration and the least organ DOX signal when normalized against tumor signal (Fig. 7b). Such localized delivery can be attributed to the penetration ability of TH-DOX to reach subcutaneous tumor region. In the case of other carriers (i.e. DS-DOX, PLGA-DOX, and LIP-DOX) and tDOX which did not show significant penetration, DOX likely had accumulated over time within the skin layers instead. Consequently, greater redistribution of free DOX is expected through the skin lymphatic or blood vessels. Hematoxylin & eosin staining corroborates the safety of FNA carrier, with minimal differences seen between organs from TH-DOX-treated mice and those from the healthy mice (Supporting Figure 15). Another striking observation is related to the utilization of PMMA MN to disrupt the integrity of skin overlying the tumor. During early timepoints (i.e. day 5/7 post tumor injection), MN application results in reversible mark which faded after 5 min. Towards the later timepoint however (e.g. day 12), thumb-pressed MN resulted in significant bleeding, presumably due to thinner skin tissue from the stretching pressure exerted (Supporting Figure 16). To this end, MN application can be considered to have induced consequential damage.

## Discussion

Transdermal penetration ability of drug carriers is greatly dependent on its physical parameters, including size, morphology, charge, and material composition. Skin penetration has so far

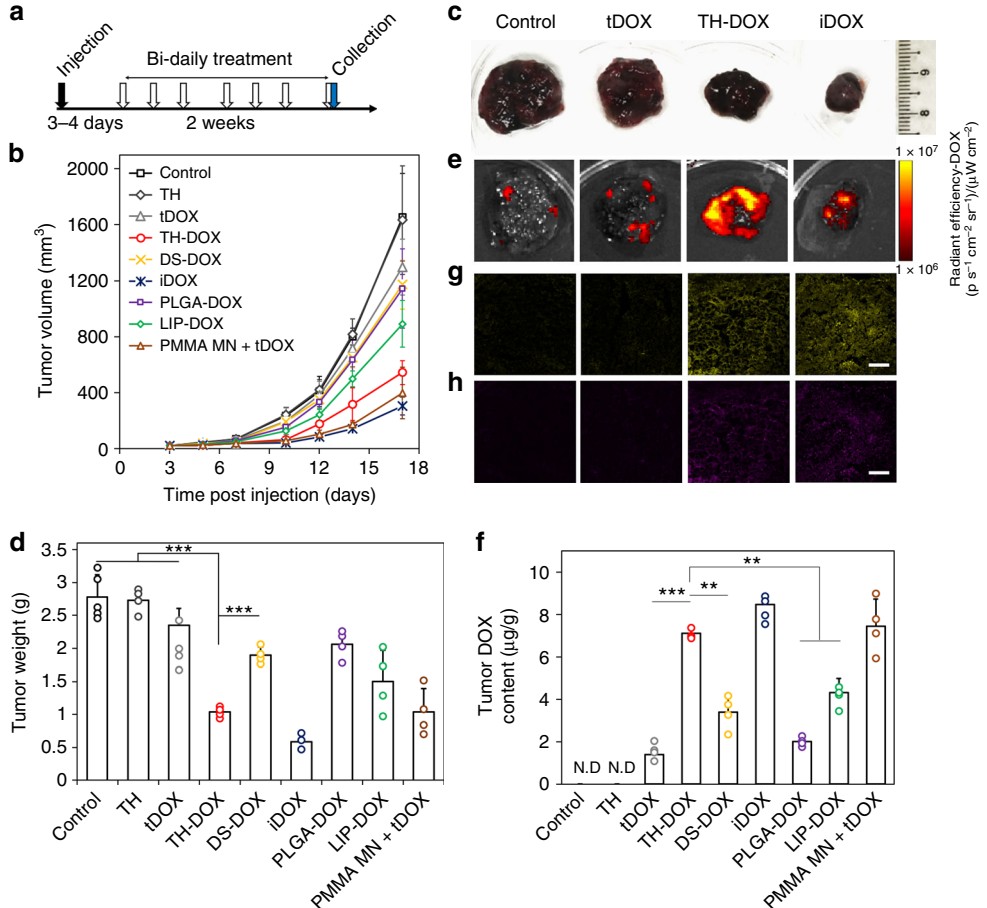

**Fig. 6** Inhibition of tumor growth with various DOX formulations in melanoma mouse model: **a** Schematic showing experimental procedure. Bi-daily treatment started 3/4 days post injection and performed for 2 weeks till sacrifice. **b** Tumor size progression in mice treated with different DOX formulations. **c** Representative photo, **d** weight and **e** IVIS imaging for DOX channel of isolated tumor samples post control, tDOX, TH-DOX, and iDOX treatments at day 17. **f** Quantification of tumoral DOX content following tumor homogenization with generated fluorescence standard curve. Each dot on bar charts represent independent sample measurement. Histological analysis of tumors in **c** for **g** Dox fluorescence (yellow channel) and **h** Annexin V expression (purple channel), scale bar: 100 μm. Source data are provided as a Source Data file. Error bars represent standard deviation. N.D: non-determinable, **$p < 0.01$, ***$p < 0.001$ as compared to TH-DOX (one-way ANOVA)

been observed on small liposomes, dendrimers, lipid and polymeric NPs and nanoemulsions, with or without adjunctive treatments (e.g. ultrasound). Predominant-positive charge promotes transdermal permeation of polymeric NPs[28], though it may also limit permeation by promoting significant adsorption on top few cell layers[29]. In this study, we find that highly ordered FNAs can penetrate the skin barrier (Fig. 2). Three structures with <75 nm in size (TH21, TH37, 6H714) showed considerable skin penetration, in which small tetrahedron TH37 (44 nm) and TH21 (17 nm) reached deeply to ~350–400 μm beneath skin surface in mice. This result aligns well with reported size-dependency on carrier skin penetrative ability[30]. However, the three distinct-shaped structures with similar size (~180 nm; TH337, R13730, and T14498) all showed minimal skin penetration (~50–75 μm from the skin surface). This suggests minimal shape dependency on FNA skin penetration, minor as compared to size dependency.

One challenge in evaluating skin penetration of wide variety of nanostructures is the choice of dosage. Herein, there is a big discrepancy in molecular weight of the 8 structures, ranging between 81.9 kDa and 4711.9 kDa. In this case, fixing same FNA mole concentration would result in 58-fold greater amount of NA to be internalized between largest structure (T/B14498) and

smallest structure (T21). Considering this, same mass concentration (i.e. equivalent NA content) was chosen for the skin penetration tests (100 μg/mL). We realize that different molar concentration gradients may result in distinct diffusion rates. Thus, 24 h were given in all experiments to ensure sufficient diffusion of all 8 FNAs. Multiple fluorophores were also conjugated on larger FNAs while small structures (TH21, TH37, and 6H714) only had a single fluorophore (Supporting Table 1). An interesting observation is that the ratio of fluorophores didn't match the ratio of mass. The addition of 2 or 3 fluorophores were sufficient to enhance the fluorescence of the larger FNAs to within an order. We suspect the 3D structure of FNA and the location of fluorophore tagging influence the relation between fluorophore concentration and fluorescence yield, which is distinct from a fluorophore solution.

In ensuring successful drug delivery, sufficient integrity/stability of the drug carrier is required throughout the administration and distribution. By examining the fluorescence change of a dual tagged, symmetrical tetrahedron structure (TH') during its skin penetration process, we demonstrated the retention of integrity of FNA structures during topical application and skin penetration steps (Fig. 3). This is supported with ~26-h half-life of FNA under 10% serum condition, suggesting an effective carrier period of

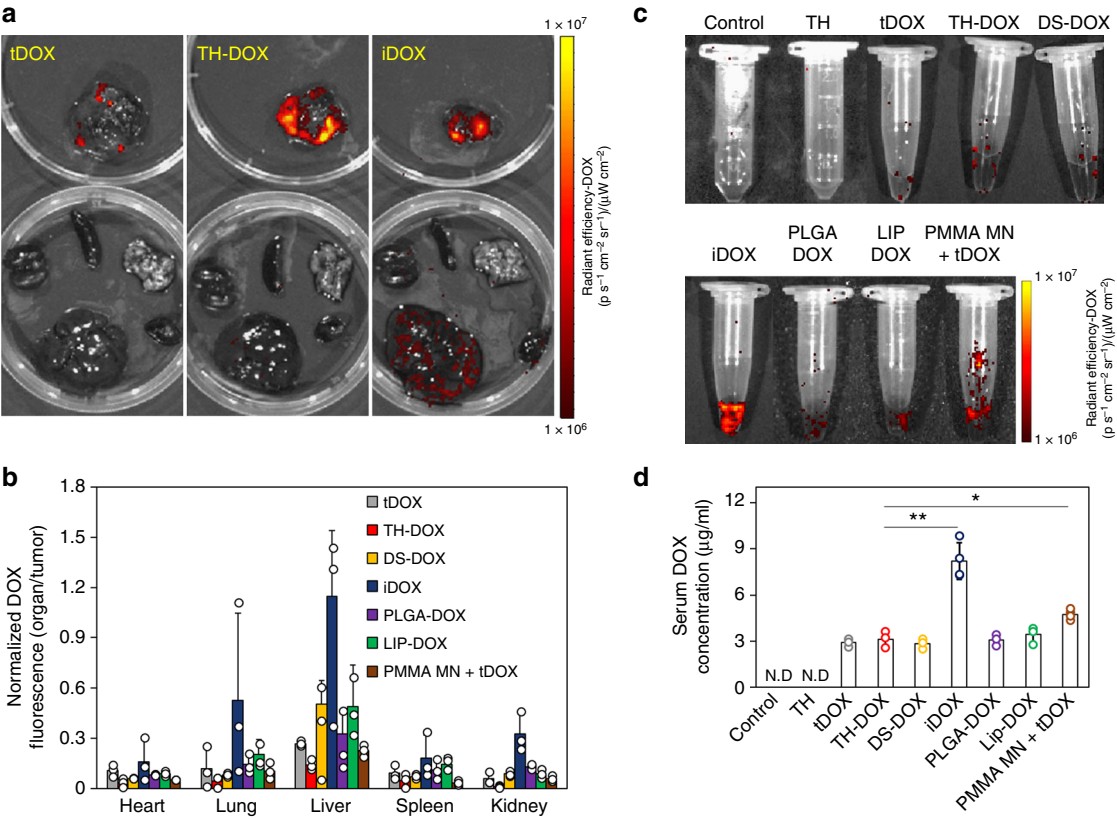

**Fig. 7** Biodistribution of DOX following treatment with different DOX formulations: **a** Representative image for tDOX, TH-DOX, and iDOX treatment group and **b** quantified DOX fluorescence from IVIS imaging of isolated tumors (top) and organs (bottom; heart, spleen, lung, liver, and kidney). **c** Representative IVIS image showing DOX fluorescence from serums of treated mice. **d** Serum DOX quantification through fluorescence fitting against standard curve. Each dot on bar charts represent an independent sample measurement. Source data are provided as a Source Data file. Error bars represent standard deviation. N.D: non-determinable value, $*p < 0.05$, $**p < 0.01$ (one-way ANOVA)

1–2 days (Supporting Figure 10). The stability of FNAs and their capability in skin penetration allowed us to explore their potential applications in skin disease treatment. Relative to conventional TDD carrier in PLGA NP and liposomes, FNAs confer physical stability with regards to aggregation and drug leakage, for inter-calated DOX tested here and expectedly for chemically con-jugated drugs (Fig. 4)[31,32]. DOX-intercalated FNA exhibited a half-life of >48 h, significantly longer than prior to DOX loading (26 h). Such stability complements its precisely tunable size and penetration ability to achieve consistent, effective TDD. Nuclease-resistant backbone modification (e.g. 2-O'-methylated), salt-modulated concentration of cations (e.g. $Mg^{2+}$), and inclusion of additional coating (e.g. oligolysine) can further stabilize FNA and extend its lifespan for even less frequent application. This will open broader range of TDD treatments, including those requiring extended treatment (e.g. chronic wound), although its influence over FNA interactions with skin cells and thus penetration requires further clarification[33–35].

Taking the xenograft melanoma tumor model as an example, capability of TH21 to deliver DOX into subcutaneous tumor site was compared to other DOX formulations (i.e. free, PLGA NPs, Liposomes, and MNs). Reaching as deep as 400–450 μm from the skin periphery (Fig. 5), TH-DOX effectively reduced the tumor growth (~33% of untreated ones), slightly less than those treated with intra-tumoral injection of DOX or with topical DOX application following PMMA MN disruption (Fig. 6b). In com-parison, DS-DOX did not penetrate deep from skin periphery (likely due to receptor interaction and premature degradation),

resulting in poor tumor growth inhibition. Similarly, inferior penetration of both PLGA-DOX and LIP-DOX as compared to TH-DOX was reflected as well in unsatisfactory tumor inhibition. More excitingly, this performance was accompanied with mini-mal non-targeted delivery on other organs. This contrasts with the great accumulation of DOX in other organs following intra-tumoral injection (Fig. 7b, c), in which locally injected DOX easily migrated through the tumor vasculature and lymph system[36]. Overall, although transdermal delivery of TH-DOX was relatively slow compared to when skin barrier was disrupted by MN application or when DOX was injected directly, better isolation of transdermal TH-DOX from the circulation enabled similar inhibition effect as the latter two. Additionally, there is minimal risk of pain and injury, unlike needle injection or microneedle application under special conditions (e.g. skin thinning)[37].

Another key advantage of FNAs as TDD carrier is its fully-biocompatible/biodegradable nature with safe degradation products. This contrasts with polymeric NPs (e.g. PLGA), inert NPs (e.g. Ag, Au) or metal-oxide NPs (e.g. ZnO, $TiO_2$ NPs) adopted in various cosmeceutical skin products, which may degrade into acids and affect physiological pH, accumulate over extended usage or release inflammatory-inducing ions[38,39]. Minimal skin irritation and better patient compliance can be expected from frequent, repeated application of FNA-based products. Finally, with its predictable hybridization and folding, FNAs of various shapes and sizes can be conveniently fabricated[1,40]. Such versatility allows careful tuning of structures to match intended delivery site/depth. As shown herein, larger FNAs such as 6H14498 and R13730 could be employed as

sustained-release drug carrier when minimal transdermal penetration is desired. Meanwhile, smaller FNAs such as TH21 can be adapted to target dermis layer (e.g. to modulate collagen production by fibroblast cells)[41].

To our knowledge, this is the first study to explore DNA origami nanostructures for TDD application. Preliminary study reveals that both cellular and trans-appendageal route contribute to DNA nanostructure skin penetration, with the latter mechanism responsible for ~75% delivery into deep dermis region. Future study will elaborate on this penetration mechanism, with emphasis on the influence of FNA size and shape and surface chemistry. Fine-tuning of FNA dosage and formulation is also crucial to optimize delivery profile, especially for translation towards more robust, denser human skin. Finally, it would be interesting to utilize these FNAs to topically deliver other drug moieties such as antisense oligonucleotides, peptides, or non-intercalating small molecules.

## Methods

**Materials**. Chemicals and reagents were obtained from Sigma-Aldrich unless stated otherwise. Dulbecco's modified eagle medium (DMEM) containing L-glutamine was purchased from Lonza. Trypsin–EDTA (0.05%), fetal bovine serum (FBS) and penicillin–streptomycin (PS; 10 kU/mL) were obtained from Gibco. NucBlue Live Ready Probes and LysoTracker® were obtained from ThermoFisher Scientific. Aquaphor Healing Ointment (Eucerin) was obtained from pharmacy. Animal studies were performed in compliance with guidelines set by Nanyang Technological University's Institutional Animal Care and Use Committee (IACUCs: NTU #BN16098).

**Fabrication and characterization**. FNAs were assembled in accordance to previous studies[42–44]. Detailed assembly of each FNA structure can be found in supporting information. Briefly, small FNAs were assembled using equimolar ratio of ssDNA component strands. Meanwhile, large FNAs were assembled by mixing together long ssDNA scaffold with short staple oligonucleotides (designed to fold different parts of scaffold DNA) at 1:10 ratio (excess of short strands) in 1X Tris-Acetate-EDTA (TAE) buffer containing MgCl$_2$ and other constituents as listed in Supporting Table 1. Sequences for FNA structures are as listed in Supporting Table 2–Supporting Table 10. DNA strands were then annealed from 95 °C to 20 °C in thermal cycler PCR machine (Eppendorf) with rate of 1 °C/min. Finally, DNA origami structures were purified using size-exclusion columns.

These structures, and partially assembled or unassembled TH21 were analyzed using gel electrophoresis and AFM. In short, 8% Polyacrylamide Gel Electrophoresis (PAGE) or 1% Agarose Gel Electrophoresis (AGE) was performed in 1X TAE buffer (4 mM Tris base, 2 mM acetic acid, 0.2 mM EDTA) including 1.25 mM Mg(CH$_3$COO)$_2$ at 4 °C for about 1~ 1.5 h. Then gel was stained with GelRed for ~15 min for evaluation purposes. For AFM measurement, origami sample (5 µl) was left to adsorb on mica surface. 30 µl 1X TAE buffer was added prior to sample scanning under tapping mode using a J scanner from Multi-mode Nanoscope IIIa AFM (Vecco/Digital Instruments), coupled with a silicon nitride cantilever with sharp pyramidal tip (OMCL-TR400PSA, Olympus). Hydrodynamic size was analyzed with Zetasizer nano Z (Malvern) by dispersing 1 µg of FNAs in 10 ml distilled water. Measurement was repeated three times.

Doxorubicin (DOX) was intercalated into TH21 and 21 base-pair dsDNA segments (DS21, TATTCTACTTGAGAGAGCGAC and GTCGCTCTCTCAA GTAGAATA (5' to 3')) via simple incubation at 10 µM concentration for 1 h to form TH-DOX and DS-DOX. Excess DOX was removed through high-performance liquid chromatography (HPLC)[5]. TH-DOX and DS-DOX were then taken for gel electrophoresis and fluorescence measurement, to confirm and quantify DOX loading. DOX-loaded PLGA NPs and liposomes were prepared according to previous protocols with slight alterations[45–47]. Five micrograms of PLGA (50:50, MW: 30–60 kDA) and 50 µg DOX were dissolved in 1 ml acetonitrile, before being added dropwise into 20 ml DI H$_2$O stirred at 800 rpm. Mixture was left stirring for 4 h to thoroughly remove acetone, and nano-precipitated PLGA NP was concentrated under reduced pressure using rotary evaporator (Buchi R-200) until a concentration of 10 mg/ml. Thin lipid film for liposomes was formed by dissolving 1,2-dipalmitoyl-sn-glycero-3-phosphocholine (DPPC), cholesterol and 1,2-distearoyl-sn-glycero-3-phosphoethanolamine-N-[amino(polyethylene glycol)-2000] (DSPE-PEG2000) in 100% ethanol (weight ratio of 65:30:5) and evaporating solvent under reduced pressure (rotary evaporator Buchi R-200). Following which, 5 mg of film was hydrated with 0.3 M sodium citrate (pH 4). After the liposomes underwent size extrusion (×10, 100 nm pore size), pH of liposome solution was adjusted to 7.5 with sodium bicarbonate. Finally, DOX was loaded through incubation (final concentration of 100 µg/ml) for 1 h at 60 °C. In both cases, unencapsulated DOX was removed through dialysis (MWCO: 3 kDa). Hydrodynamic size of PLGA-DOX and LIP-DOX was analyzed with Zetasizer nano Z (Malvern) by dispersing 1 mg particle in 10 ml distilled water. Measurement was repeated three times.

**Cell culture and labeling with FNAs**. Normal dermal fibroblasts (NDF, CellResearch Corporation Pte Ltd, Singapore) and immortalized keratinocytes (HaCat, ATCC) were cultured in high-glucose DMEM (4500 mg/L) with 4 mM L-glutamine, 10% FBS and 1% PS at humidified condition of 37 °C, 5% CO$_2$. Fresh culture medium was added every 2–3 days. For labeling purpose, FNAs were added into culture medium to a final concentration of 0.2 µg/ml and left for 24 h incubation, except otherwise stated. Excess FNAs were rinsed thrice with PBS, before cellular fluorescence imaging or Alamarblue assay. In accordance to manufacturer's protocol, Alamarblue reagent was added in 1:100 volume ratio to culture medium, and left incubated for 8 h prior to fluorescence measurement (570/585 nm) to measure cell viability.

**Topical application of FNAs on mice and ex vivo human skin**. Animal studies were performed in compliance with guidelines set by Nanyang Technological University's Institutional Animal Care and Use Committee (IACUCs: NTU #BN16098). Here, six-week-old NCR nude mice were used for all the in vivo experiments. Ex vivo human skin explants were obtained from CellResearch Corporation Pte Ltd, Singapore. For topical application, 5 µl FNAs (100 µg/ml) was physically mixed with Aquaphor ® (1:1 weight ratio) until it reached homogenous consistency. Cream formulation was then applied evenly, covered with Tegaderm™ dressing and left to penetrate the skin for 24 h. After wiping excess cream, IVIS® Spectrum CT (PerkinElmer, Singapore Pte Ltd) was used to evaluate total penetrated FNAs. Fluorescence readings of region-of-interest (ROI) at respective channels (cy3: 550/570 nm, cy5: 650/670 nm, cy5.5: 675/695 nm) were taken using the Living Image 4.0 Software and normalized against readings from untreated skin.

**Application of DOX formulations on ex vivo porcine skin**. Fresh porcine ear skin was obtained from Jurong Abattoir (Singapore) and cut into 1 cm by 1 cm pieces. Ten microliter of free DOX, TH-DOX, DS-DOX, PLGA-DOX, and LIP-DOX solutions (adjusted to contain 5 µg of DOX) was mixed with Aquaphor ® (1:1 weight ratio) and topically applied for 24 h. As control, 600 µm-tall PMMA MN (3 M) was thumb-pressed on skin for 1 min prior to free DOX application. Excess cream was wiped for further processing.

**Melanoma mice model**. IC$_{50}$ of TH-DOX formulation was tested on B16F10 mice melanoma cells, by performing Alamarblue assay 72 h post treatment. Melanoma disease model was prepared on NCR nude mice or C57Bl6 mice through subcutaneous injection of $1 \times 10^6$ B16F10 melanoma cells into its rear flank. Cells were injected in 1:1 PBS: matrigel mixture to ensure retention. Incubation was given to allow tumor to reach ~50 mm$^3$ in volume prior to any treatment. Once the tumor reached appropriate size, topical application of free DOX, TH-DOX, DS-DOX, PLGA-DOX, LIP-DOX or injection of free DOX was performed to deliver ~2.4 µg DOX each treatment. This matches previous reported topical dosage of 120 µg/kg DOX[48]. Following 24-h treatment period, similar IVIS procedure was done to measure total DOX and TH21 origami at tumor region (DOX: 490/595 nm, cy5.5: 675/695 nm). For efficacy evaluation, the study was extended for another 2 weeks, with treatments repeated thrice a week. As control, several tumors were treated with unloaded TH or thumb-pressed for 1 min with PMMA MN prior to free DOX application. Tumor sizes were recorded every other day, with tumor and organs isolation following mice euthanasia. The tumor volume was calculated by the following formula: length × width$^2$/2[49]. For serum DOX quantification, mice blood samples were centrifuged at 4000×$g$ for 15 min after 30 min incubation at room temperature.

**Histology**. For histological examination, mice and human skin explants after topical FNA treatment were fixed using 10% neutral-buffered formalin solution for 24 h. Skins were subsequently rinsed thrice with PBS, treated with 30% (w/v) sucrose for 16 h, and immersed into optimal cutting temperature (OCT) solution. Samples were then exposed to liquid nitrogen for cryosectioning. Finally, thin sample slices (15 µm) were stained with Hoechst33342, Annexin V AlexaFluor647 (Invitrogen™ A23204, 50-fold dilution) or anti-Keratin14 antibody (DyLight 488; Novus Biologicals NBP2-34675G, 5 µg/mL) according to the manufacturer's protocol or with standard Hematoxylin & Eosin protocol.

**Fluorescence imaging and signal quantification**. Fluorescence imaging on FNA-labeled cells or cream-treated skin sections was performed on Laser Scanning Microscope LSM800 (Zeiss) at ×100 magnification. Respective fluorescence channels (DyLight 488: 485/515 nm, cy3: 550/570 nm, cy5: 650/670 nm, cy5.5: 695/715 nm, red dnd-99: 577/590 nm, Hoechst33342: 358/461 nm, and Doxorubicin: 485/585 nm) and transmitted ESID light were kept constant for all samples throughout the experiment. ImageJ software was used to remove background fluorescence and quantify fluorescence intensity within individual cells and across skin depth. Unlabeled cells and untreated skins were used for signal normalization.

**Statistical analysis**. One-way ANOVA was carried out to calculate the significance $P$-value. A suitable post hoc test (Tukey) was chosen using the IBM SPSS Statistics 22 software. For each experiment, values are reported as mean ± standard deviation of at least three-independent samples.

**Reporting summary**. Further information on experimental design is available in the Nature Research Reporting Summary linked to this article.

## Data availability

All relevant data are included in the main manuscript and Supplementary Information. The source data underlying Fig. 2c, e, 3d, e, g, 4b-d, g, 5b, e-g, 6b, d, f, 7b, d and Supplementary Figs 1c-d, 2b, 4a-b, 5, 7b-c, 9b, 13b-c and 14b are provided as a Source Data file. Additional data are available from the corresponding author upon reasonable request.

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

## Acknowledgements

This work was supported by the Singapore A*STAR Biomedical Research Council (IAF-PP grant), National Key R&D Program of China (2016YFA0400900), National Natural Science Foundation (21834007, 21675167, 11675251). The authors thank Dr Kai Xuan Keith Tan for assistance in performing Krt14 antibody staining.

## Author contributions

C.W. and Y.Z. performed research. M.X., W.F. and Q.L. synthesized and characterize the FNAs. D.L. performed the histology. C.W., D.L. and D.C.Y. conducted animal experiments. M.Z. assisted gel electrophoresis. C.W., C.F. and C.J.X. planned the experiments, analysed the results, and drafted the manuscript. M.V.S. contributed discussion and provided clinical insights. L.W. designed and supervised the studies, and finalized the manuscript. All authors reviewed and approved the final version of the manuscript.

## Additional information

**Competing interests:** A patent application has been submitted based on these results. The patent application covers the design criteria and synthetic methods of the Framework Nucleic Acids (FNAs) for transdermal applications, with co-authors C.F., C.J.X., L.W., C.W. and M.X. listed as inventors. Application number. CN201811308594. Status of application: Publication (2019.02.22); Publication number: CN109364261A. The remaining authors declare no competing interests.

