## [Peer Review File · Nature Communications]

Reviewers' Comments:

Reviewer #1:

Remarks to the Author:

Rationally designed DNA nanostructures with programmable size, shape, dynamics, and modification are promising drug-delivery reagents. However, fundamental questions regarding the entry pathway, circulation time, biodistribution, stability, etc. have to be answered to inform the design decisions of DNA-nanostructure-based drug carriers. The field has made considerable progress in defining the DNA structure size- and shape-dependency of cellular uptake efficiency as well as demonstrating the utility of DNA carriers for target-specific drug release. Most previous studies deliver DNA structures to model animal via injection (e.g. intravenous, intratumoral). This manuscript describes the delivery of DNA structures in a less invasive, more targeted, and potentially safer way, namely by transdermal delivery. The delivery efficiencies of eight different DNA constructs were quantified using cultured cells and nude mice. The delivery pathway, in vivo stability, dox-loading efficiency, delivery efficacy, and biodistribution of the best-performing carriers, typically <70 nm tetrahedrons, were systematically studied. This study is therefore very informative for people interested in the therapeutic application of DNA nanotechnology. The manuscript is written with appropriate background for the general readership of Nature Communications and sufficient depth for the experts in nanomedicine.

I have the following questions that I hope the authors to answer:

(1) The transdermal delivery of DOX seems like a "niche" application that are only good for treating a limited number of unique diseases. While I understand (and appreciate) the value of the proof-of-concept demonstrations presented here, it would be nice if the authors clearly define the limitation of the current techniques. For example, what diseases could directly benefit from FNA delivery? what are technical hurdles one needs to overcome in order to make the FNA carriers more broadly applicable?

(2) What limits the penetration depth in this study? Is there room for further improvement? For example, if the FNA structural integrity is the limiting factor, what can be done to improve that in the future (e.g. chemical modification)?

(3) How does the delivery using small FNAs comparing to using simple dsDNA segments? If the size of DNA structure is the determining factor of penetration depth, wouldn't one expect short dsDNA to work equally well, if not better than the small FNAs?

(4) The fluorescence labeling scheme is unclear. Authors seem to want to compensate the larger DNA structures with more fluorophores when they normalize dosage by mass. But from the schematics in Fig S1 this is hardly quantitative. For example, structures (5-8) with ~20-60 times larger M.W. than structures (1,2,and 4) are only labeled with 3-4 times as many fluorophores. Yet, Fig S5 shows that at the same mass/volume concentration, these FNAs have similar fluorescence level. This seems to indicate problems in sample purity, labeling efficiency and/or concentration measurement.

(5) No detailed design blueprints are provided for the DNA structures, which are essential for anyone who wants to replicate the experiments.

(6) The FRET-based stability tests, while nicely designed and executed, technically only prove the local stability of one edge of the tetrahedron. An argument could be made that if all edges are designed in the same way they should have similar stability. However, the detailed design diagram is missing... A more religious study would be to build six permutations of the tetrahedron each with a different edge labeled with a dye-quencher pair and compare their stability in tissue cultures and ex vivo skins.

(7) Recent reports outlines the effect of DNA structure size and shape in cellular uptake should be cited. (10.1021/jacs.7b09024, 10.1021/acs.nanolett.8b00660)

Reviewer #2:

Remarks to the Author:

The authors describe in this paper the exploration of several DNA "origamis" for their potential to penetrate skin and to deliver with doxorubicin a DNA-intercalating drug to a melanoma model. Although potentially interesting, the current version of the manuscript suffers of several experimental shortcomings.

1. The preservation of structural integrity is an important component for the suggested FNA delivery approach. However, the authors provide only indirect data addressing this question. For instance, what is the half-life of TH21 in 10% FCS? Is this affected by doxorubicin? Simple gel electrophoresis experiments would already provide some important insight here.
2. The data shown in Figs 2 and S1-S3 suggest a correlation between hydrodynamic diameter and penetration efficacy. How does the origami compare to a much simpler structure such as a simple 21 bp double strand that would also be able to accommodate doxorubicin? This very important and potentially informative control is completely missing and is absolutely required for understanding the impact of DNA structure for transdermal drug delivery.
3. Poor performance of the liposomal dox control is not surprising giving the absence of a pH gradient, which would reduce dox leakage, and the apparent lack of extrusion to control for liposome size.
4. The impact of the different dox formulations on tumour progression (Figure 6) is interesting, but incompletely controlled. What is the impact of TH21 alone on tumour growth?

Minor points:

5. How were FNAs tagged with fluorophores?
6. In general, the figure legends lack crucial information to understand the figures. For instance, what are the dotted lines in figure 4d, what is shown in 4e?
7. The nuclear counter stain is Hoechst 33342, not Hoescht.

(Remarks to the Author) Reviewer #1:

Rationally designed DNA nanostructures with programmable size, shape, dynamics, and modification are promising drug-delivery reagents. However, fundamental questions regarding the entry pathway, circulation time, biodistribution, stability, etc. have to be answered to inform the design decisions of DNA-nanostructure-based drug carriers. The field has made considerable progress in defining the DNA structure size- and shape-dependency of cellular uptake efficiency as well as demonstrating the utility of DNA carriers for target-specific drug release. Most previous studies deliver DNA structures to model animal via injection (e.g. intravenous, intratumoral). This manuscript describes the delivery of DNA structures in a less invasive, more targeted, and potentially safer way, namely by transdermal delivery. The delivery efficiencies of eight different DNA constructs were quantified using cultured cells and nude mice. The delivery pathway, in vivo stability, dox-loading efficiency, delivery efficacy, and biodistribution of the best-performing carriers, typically <70 nm tetrahedrons, were systematically studied. This study is therefore very informative for people interested in the therapeutic application of DNA nanotechnology. The manuscript is written with appropriate background for the general readership of Nature Communications and sufficient depth for the experts in nanomedicine.

Response: We fully agree with the reviewer's remark regarding the potential DNA structures for the non-injection-based drug delivery (including transdermal delivery), which has motivated our study presented herewith. We sincerely appreciate the constructive input the reviewer has provided to improve our manuscript. We have performed additional experiments and revised the manuscript based on the reviewer's remarks.

I have the following questions that I hope the authors to answer:

(1) The transdermal delivery of DOX seems like a "niche" application that are only good for treating a limited number of unique diseases. While I understand (and appreciate) the value of the proof-of-concept demonstrations presented here, it would be nice if the authors clearly define the limitation of the current techniques. For example, what diseases could directly benefit from FNA delivery? what are technical hurdles one needs to overcome in order to make the FNA carriers more broadly applicable?

Response: We appreciate the critical opinion of the reviewer about the limitations and further improvement of this platform technology. Noting its 1-2 days structural integrity and inherent compatibility with Nucleic Acid therapeutics (e.g. RNA interference, antisense oligonucleotides), we believe FNA transdermal delivery is suited for skin diseases which necessitate localized treatment and gene expression modulation. Structural integrity of FNA is a crucial hurdle to take into consideration for prolonging its lifespan, enabling less frequent application and thus opening broader range of TDD treatments. We have amended this information in the introduction (page 4) and discussion (page 14) section of the manuscript.

(2) What limits the penetration depth in this study? Is there room for further improvement? For example, if the FNA structural integrity is the limiting factor, what can be done to improve that in the future (e.g. chemical modification)?

Response: From our observation, size plays the most crucial role in dictating diffusional rate and extent of skin penetration. FNA of size >70nm showed minimal penetration (Figure 2). On the other hand, very small DNA structures (<10 nm, in the case for DS21) similarly exhibited low penetration (Figure 4F-G). In this regard, as reviewer pointed out, we believe structural integrity is one crucial aspect impacting its penetration. Premature degradation would cause FNAs to be retained in initial few cell layers, unable to penetrate deeper. Consequently, we agree that chemical modification is beneficial in improving FNA efficacy in TDD. Additionally, polymer coating and additional chemical linker (e.g. oligolysine) could further stabilize FNA integrity, although its influence over FNA interactions with skin cells and thus penetration requires further evaluations. This information has been amended into the discussion section of the manuscript (page 14).

(3) How does the delivery using small FNAs comparing to using simple dsDNA segments? If the size of DNA structure is the determining factor of penetration depth, wouldn't one expect short dsDNA to work equally well, if not better than the small FNAs?

Response: We thank the reviewer for pointing out this critical aspect of FNA delivery. Accordingly, we have performed skin penetration and mouse tumour treatment using small 21 base-pair dsDNA (DS21). As shown in fig

4F&G, its skin penetration ability was much weaker than TH21, but comparable to liposome formulation. Consequently, its tumor inhibition ability was weaker than TH21 as well (fig 6B&D, fig S12-S14), despite having comparable DOX loading capacity to TH21 (fig 4A). Likely, such small DNA constructs are retained by the initial few cell layers, through interaction with membrane receptors. Coupled with its susceptibility to degradation, they consequently were unable to maintain integrity to penetrate and reach deeper skin layers. These results and related discussion have been included in subsection 'Transdermal drug delivery with topical FNAs' (page 9), subsection 'Efficacy of topical FNAs for cancer treatment' (page 11), and the discussion section (page 14).

(4) The fluorescence labelling scheme is unclear. Authors seem to want to compensate the larger DNA structures with more fluorophores when they normalize dosage by mass. But from the schematics in Fig S1 this is hardly quantitative. For example, structures (5-8) with ~20-60 times larger M.W. than structures (1,2, and 4) are only labelled with 3-4 times as many fluorophores. Yet, Fig S5 shows that at the same mass/volume concentration, these FNAs have similar fluorescence level. This seems to indicate problems in sample purity, labelling efficiency and/or concentration measurement.

Response: We appreciate the sharp observation of the reviewer. Our 8 FNA structures are tagged with Cy5.5 fluorophore(s) through hybridization of Cy5.5-labeled short strand (table S2-S9), which were added in excess (10:1) ratio in larger structures (TH337, R13730, etc.) to ensure successful tagging at designated locations. Subsequently, assembled FNAs were purified thrice by running through HPLC or centrifugation filters. Through this process, un-assembled staple and Cy5.5 strands were removed from the assembled FNAs. Nevertheless, the number of fluorophores per FNA was one of the big challenges we faced during the design of the experiment. Ideally, we should match the ratio of fluorophores between different structures to the ratio of mass. However, we noted that the ratio of fluorophores didn't match the ratio of mass. The addition of 2 or 3 fluorophores were sufficient to enhance the fluorescence of the larger FNAs to within an order. We suspect the 3D structure of FNA and the location of fluorophore tagging influence the relation between fluorophore concentration and fluorescence yield, which is distinct from a fluorophore solution. This information has been included in 'Topical delivery of FNAs in mouse and human skin explant' (page 6) and the discussion section (page 13).

(5) No detailed design blueprints are provided for the DNA structures, which are essential for anyone who wants to replicate the experiments.

Response: We have included the detailed blueprints in supporting information, table S2-S10. Procedures for FNA assembly and fluorescence tagging have also been described in supporting information.

(6) The FRET-based stability tests, while nicely designed and executed, technically only prove the local stability of one edge of the tetrahedron. An argument could be made that if all edges are designed in the same way they should have similar stability. However, the detailed design diagram is missing... A more religious study would be to build six permutations of the tetrahedron each with a different edge labeled with a dye-quencher pair and compare their stability in tissue cultures and ex vivo skins.

Response: We appreciate the reviewer's suggestion to rigorously study the FNA stability with tetrahedron as an example. The detailed sequences and protocols for FNA assembly have been included in supporting information (table S2-S10). Considering equivalent C/G and A/T ratio of each edge, we believe the stability of one TH edge sufficiently represents the stability of all the edges. This is further supported with gel electrophoresis study showing similar distance travelled for all structure permutations containing same number of ssDNA components (i.e. missing 2 chains: 2A-2F, missing 1 chain: 3A-3D; fig 3A&B). Additionally, it is to be noted that a third experimental group in incomplete TH (i.e. missing one strand to simulate TH' disassembly; i-TH') have been shown to exhibit comparable Cy5 fluorescence as in unquenched TH (fig 3C&D). This information has been added into the subsection 'Topical delivery of FNAs in mouse and human skin explant' (page 7).

(7) Recent reports outlines the effect of DNA structure size and shape in cellular uptake should be cited. (10.1021/jacs.7b09024, 10.1021/acs.nanolett.8b00660)

Response: We have included these recent reports which outline the effect of size and shape for cellular uptake of DNA structures in the revised manuscript "This aligns well with previous report... dictate how well it interacts with the cell receptors" ('In vitro characterization of FNAs' (page 5); reference no 18 & 19).

(Remarks to the Author) Reviewer #2:

The authors describe in this paper the exploration of several DNA "origamis" for their potential to penetrate skin and to deliver with doxorubicin a DNA-intercalating drug to a melanoma model. Although potentially interesting, the current version of the manuscript suffers of several experimental shortcomings.

Response: We thank the reviewer for the positive remarks on DNA origami for transdermal drug delivery. We sincerely appreciate the constructive input the reviewer has provided to improve our manuscript, and we have performed more experiments and revised the manuscript based on the reviewer's remarks.

1. The preservation of structural integrity is an important component for the suggested FNA delivery approach. However, the authors provide only indirect data addressing this question. For instance, what is the half-life of TH21 in 10% FCS? Is this affected by doxorubicin? Simple gel electrophoresis experiments would already provide some important insight here.

Response: We fully agree that structural integrity is a key aspect of FNA drug delivery and have studied as suggested TH21 stability in 37°C, 10% serum condition through gel electrophoresis. As shown in figure S10, the half-life of TH21 and Cy5.5-tagged TH21 is comparable at 26 and 32h, respectively. Therefore, TH21 delivery can be expected to remain effective within 1-2 days. The presence of intercalating doxorubicin indeed stabilized the TH21 structure, with significantly extended half-life beyond 48h. Thus, TH-DOX formulation can remain efficacious for even longer post skin penetration. This information has been added into the subsection 'Transdermal drug delivery with topical FNAs' (page 8) and the discussion section (page 13).

2. The data shown in Figs 2 and S1-S3 suggest a correlation between hydrodynamic diameter and penetration efficacy. How does the origami compare to a much simpler structure such as a simple 21 bp double strand that would also be able to accommodate doxorubicin? This very important and potentially informative control is completely missing and is absolutely required for understanding the impact of DNA structure for transdermal drug delivery.

Response: We thank the reviewer for this great suggestion. Accordingly, we have loaded DOX, performed skin penetration and in vivo mice tumor treatment using small 21 base-pair dsDNA (DS21). While it exhibited similar DOX loading capacity as TH21 (fig 4A), DS21 showed significantly lesser skin penetration than TH21 (fig 4F&G). Consequently, it exhibited poor inhibition of mice tumor growth (fig 6, fig S12-S14). Likely, such small DNA constructs were retained by the initial few cell layers, through interaction with membrane receptors. Coupled with its susceptibility to degradation, they consequently are unable to maintain integrity to penetrate and reach deeper skin layers. These results have been included in subsection 'Transdermal drug delivery with topical FNAs' (page 9), subsection 'Efficacy of topical FNAs for cancer treatment' (page 11) and the discussion section (page 14).

3. Poor performance of the liposomal dox control is not surprising giving the absence of a pH gradient, which would reduce dox leakage, and the apparent lack of extrusion to control for liposome size.

Response: We appreciate the fantastic suggestion on using pH-gradient liposome loading/extrusion size control that we overlooked. We have repeated all the related with experiment and have updated the old LIP-DOX data. Briefly, we noted greater loading and DOX retention following 4/7 days storage, accompanied with slightly less aggregation (fig 4A, C & D). We subsequently studied its ex vivo penetration (fig 4F, G) and in vivo tumor growth inhibition (fig 6, fig S12-S14), which showed slightly improved results from the original LIP-DOX. Nonetheless, FNA formulation in TH-DOX still demonstrated better tumor growth inhibition. This is in agreement with greater skin penetration ability of TH-DOX (fig 4F, G). These results have been included in subsection 'Transdermal drug delivery with topical FNAs' (page 9) and the subsection 'Efficacy of topical FNAs for cancer treatment' (page 11).

4. The impact of the different dox formulations on tumor progression (Figure 6) is interesting, but incompletely controlled. What is the impact of TH21 alone on tumor growth?

Response: We thank the reviewer for pointing out this missing control. We actually had performed the experiment but did not include it, given that the focus of this article is to compare it with other formulations. We have now updated the manuscript with the *in vivo* tumor growth results from treatment with TH21 alone. As shown in fig 6 and fig S12-S14, TH21 by itself did not impede tumor growth, as compared to control treatment. This aligns well with the low cytotoxicity DNA origami constructs posed on fibroblasts and keratinocytes (fig S4B), corroborating the great biocompatibility of FNA as drug carrier. This result has been included in subsection 'Efficacy of topical FNAs for cancer treatment' (page 11).

5. How were FNAs tagged with fluorophores?

Response: We have included the detailed blueprints in supporting information, table S2-S10. Complete procedures for FNA assembly and fluorescence tagging have also been described in supporting information. Briefly, designed DNA

strands, staple strands and complementary fluorophore strand were mixed in corresponding buffer solution and annealed under suitable cooling rate.

6. In general, the figure legends lack crucial information to understand the figures. For instance, what are the dotted lines in figure 4d, what is shown in 4e?

Response: We have checked and added details on the figure legend, especially for figure 4. Briefly, dotted grey lines in fig 4D show the size distribution of TDD carrier at day 7 upon 4⁰C storage, whereas the continuous black lines show its sizes right after fabrication. Figure 4E shows the morphology of PMMA MN used as comparison to mediate topical DOX delivery.

7. The nuclear counter stain is Hoechst 33342, not Hoescht.

Response: We have revised all Hoechst33342 labelling as suggested.

Reviewers' Comments:

Reviewer #1:

Remarks to the Author:

The authors have adequately addressed my concerns. The new control experiments are thorough. I support publication.

Reviewer #2:

Remarks to the Author:

The authors addressed all experimental questions to my satisfaction. However, i still find the figure legends rather cryptic with little explanation about what actually can be seen (i.e. which colour is what, very basic info about procedures, etc).

Olaf Heidenreich

(Remarks to the Author) Reviewer #1:

The authors have adequately addressed my concerns. The new control experiments are thorough. I support publication.

Response: We sincerely appreciate the constructive input the reviewer has provided to improve our experiments as well as manuscript.

(Remarks to the Author) Reviewer #2:

The authors addressed all experimental questions to my satisfaction. However, i still find the figure legends rather cryptic with little explanation about what actually can be seen (i.e. which colour is what, very basic info about procedures, etc).

Olaf Heidenreich

Response: We sincerely appreciate the constructive input the reviewer has provided to improve our manuscript. In the revised manuscript, we have included further details on the figure legends to explain color coding and procedures performed. We hope the improved figure legends are adequately clear to explain the figures.